# Factors associated with Anganwadi Workers' service delivery of Integrated Child Development Services (ICDS) in rural India: A cross-sectional analysis of household and community health worker surveys

Lakshmi Gopalakrishnan[1]*, Sumeet Patil[2], Lia Fernald[3], Dilys Walker[1], Nadia Diamond-Smith[1]

1 University of California, San Francisco, California, United States of America, 2 NEERMAN, Unit 3, Mahendra Industrial Estate, Mumbai, Maharashtra, India, 3 UC Berkeley School of Public Health, Berkeley, California, United States of America,

* Lakshmi.gopalakrishnan2@ucsf.edu

## Abstract

### Objectives

India's 1.4 million Anganwadi Workers (AWW), a type of community health worker (CHW), serve 158 million beneficiaries under the Integrated Child Development Services (ICDS) program. They play a crucial role in facilitating the delivery of nutrition services at the village level in India. Despite this, quantitative evidence regarding the factors that influence the service delivery of AWW in India is limited.

### Methods

We used data from 6653 mothers of children below 12 months, 2398 pregnant women, and 1344 AWW from 841 villages in Bihar and Madhya Pradesh collected in 2018-19. AWW performance was operationalized as product-oriented services, such as growth monitoring and supplementary food, and information-oriented services, such as the number of home visits and counseling on infant and young child feeding practices (IYCF). We fitted multivariate logistic regression models for each outcome using a set of AWW variables, applying the Means, Motives, and Opportunity (MMO) framework: means (AWW capabilities, including education and experience), motives (AWW willingness to perform, including motivation and supervision), and opportunities (AWW chance to perform, including training, AWW caste, and workload).

### Results

Regarding product-oriented services, approximately 48% of beneficiaries received growth monitoring services, 52% of women received take-home rations, and 20%

**Data availability statement:** Dataset for this paper publicly available at Harvard Dataverse: https://dataverse.harvard.edu/dataset.xhtml?persistentId=doi:10.7910/DVN/6ANTRQ.

**Funding:** This study is funded by Grant No. OPP1158231 from the Bill and Melinda Gates Foundation to the University of California, San Francisco and University of California, Berkeley. The funder (Bill and Melinda Gates Foundation) reviewed and approved the study design, but was not involved in data collection, data analysis, data interpretation, or writing of the report. The corresponding author had full access to all the data in the study and had final responsibility for the decision to submit for publication.

**Competing interests:** The authors have declared that no competing interests exist.

received hot-cooked meals. Regarding information-oriented service delivery, more than a third (37%) received home visits, and 45% of women got counseling on IYCF. Opportunity-related factors such as AWW caste, training, and availability of facilities and resources were significantly associated with the receipt of product-oriented services. For information-oriented services, motives and opportunity-related factors were significantly associated, including motivation, timely salary receipt, AWW caste, supervision, and training.

## Conclusion

Harnessing CHWs' skills and performance could address healthcare system challenges, extend program reach, and accelerate progress toward Universal Health Coverage. Our research underscored the importance of factors such as training, access to resources, and service delivery of AWW.

## Trial registration number

https://doi.org/10.1186/ISRCTN83902145.

## Introduction

Community health workers (CHW) are a critical workforce providing health and nutrition services and increasing coverage and access to essential health and nutrition interventions in rural and underserved areas across many low-and middle-income countries [1–4]. Globally, there are still substantial gaps in knowledge and evidence regarding how to effectively support CHW programs in achieving both extensive coverage and high-quality interventions [2].

India's leading nutrition program, the Integrated Child Development Services (ICDS), has nearly 1.4 million CHWs known as Anganwadi workers (AWW). Each AWW caters to a catchment area of approximately 1000 people. They operate through a network of Anganwadi Centers (AWCs), or early childhood development and feeding centers [5–7], delivering the ICDS services: growth monitoring activities for children as appropriate; delivering supplementary food including hot-cooked meals and take-home rations; conducting home visits and counseling to educate pregnant and lactating women on pregnancy care and infant and young child feeding practices (IYCF) [5,8]. The program has routine monitoring conducted by the dedicated cadre of supervisors who supervise activities performed by a cluster of 20–25 AWW through monthly monitoring visits [9,10]. AWW are selected from local villages by State-appointed committees, with minimum qualification requirements of matriculation (10th standard) and age limits of 18–35 years. As honorary workers rather than formal employees, AWW receive modest honoraria (₹4,500 monthly or 52 USD plus performance-linked incentive as of 2021), which may be supplemented by state governments. This recruitment approach directly influences both AWW capabilities and motivation – local recruitment enhances community embeddedness but potentially limits the educational

qualifications of candidates, while the honorary worker status and compensation structure may affect motivation and retention, particularly given the substantial responsibilities assigned to AWW [11].

Prior research has identified several challenges in nutrition service delivery, including poor community sensitization, deficient record-keeping by AWW, inadequate monitoring of AWW, lack of timely service delivery, insufficiently equipped centers, insufficient training, and limited supervision [12–16]. National survey data also echo the deficiencies in the system—only 67% of mothers reported receiving food supplements, growth monitoring, and pre-school education services from AWW for children below six with wide variation between states [17]. A study on ICDS using nationally representative data highlighted an improvement in usage of ICDS services between 2006 and 2016, though they also noted program's inability to effectively reach the poorest households and women with limited education, particularly in states grappling with the highest rates of undernutrition [18].

AWW can play a vital role in delivering health services to the marginalized and underserved populations in rural and remote areas. Understanding the factors associated with AWW performance or service delivery holds the potential to guide future strategies, offering a promising opportunity to improve AWW performance and subsequently women and children's health, particularly within resource-constrained contexts.

A few global systematic reviews have highlighted a broad range of macro-level health systems and individual factors that influence CHW performance [3,19–22]. Some reviews have identified multiple health-system factors, including supervision, training, incentives, availability of supplies, functional supply chains, job aids, remuneration, workload, among others, can influence CHW performance [3,21]. Other reviews noted CHW individual-level factors such as being female, having fewer household duties, better educational status, family support, social status, CHW competence as being important factors that can influence CHW performance [3,21], though the evidence is mixed on some of these factors [3]. Apart from the health systems and individual factors, contextual factors including, local economic, political systems, power dynamics, social norms, community characteristics have also been recognized as important for CHW performance [22,23].

Assessing CHW performance is context-specific [19] and yet there is a lack of quantitative research investigating the specific factors linked to AWW performance from India. Previous evidence from India has predominantly taken a qualitative approach when exploring the determinants of CHW/AWW performance [24–26]. For instance, John and colleagues (2020), in their qualitative examination of AWW performance in Bihar, one of India's poorest states, highlighted the need to consider various interconnected factors across the individual, program, community, and organizational levels. The authors found that the primary obstacles to AWW performance often originated from external factors beyond the AWWs' control, including limited program resources, caste dynamics, seasonal migration, and corruption [25].

Quantitative studies on the drivers of AWW performance are relatively limited. One study explored the quantitative predictors of AWWs' services and found that monetary incentives were associated with a higher likelihood of households receiving general nutrition information [5]. Another study focused on the impact of supportive supervision and found that a more intensive level of supportive supervision, characterized by monitoring visits and training, was linked to improved AWW performance [10]. Furthermore, a quantitative study examining AWW performance, specifically in terms of their time allocation, found a positive and significant association between AWW education and their likelihood of maintaining records [27]. However, gaps persist when it comes to investigating the constellation of individual and health system factors influencing AWW performance in the context of key services of the ICDS program. Throughout this paper, we use the terms 'performance' and 'service delivery' interchangeably. In our study, we defined performance as AWW providing services in accordance with the ICDS guidelines.

In this paper, we sought to bridge this gap by examining the health system and individual factors associated with the performance of AWW, using the adapted ***Means, Motives, and Opportunities*** framework as shown in Fig 1 [24]. The definitions of each are provided below:

1. **Means**: This domain assessed whether the AWW possessed the necessary capabilities to perform the service. We conceptualized this to include individual characteristics of AWW, including education, level of experience, content knowledge, and skills (application knowledge).

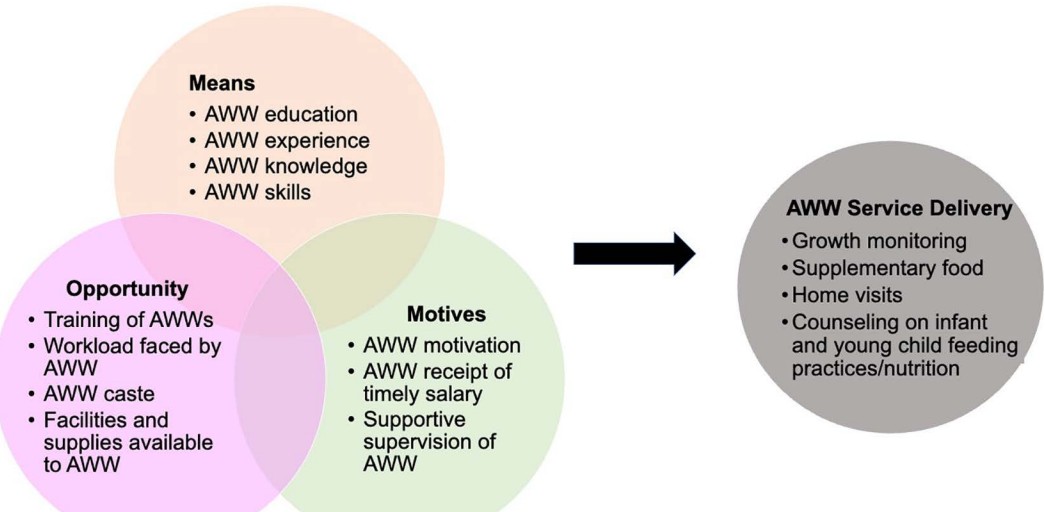

**Fig 1. Adapted Means, Motives, Opportunity Framework to Study AWW service delivery/performance [8].**

2. **Motives**: This domain explored whether the AWW was motivated to carry out the service. It included the AWW's personal motivation, financial incentives (timely payment of salary), and health system-related factors like the presence of supportive supervision.

3. **Opportunity**: This domain examined whether the AWW had the *chance or opportunity* to perform the service effectively. This includes considerations such as the availability of facilities and resources to support AWW in delivering services, especially for tasks related to growth monitoring and the distribution of supplementary food. AWW workload and AWW caste as potential factors were also included.

Following a precedent from a prior paper on AWW in India, we classified the AWW performance into two distinct categories [5]. Services involving home visits and counseling on infant and young child feeding practices were categorized as "information-oriented services" since they primarily involve the dissemination of information and education to pregnant and lactating women. In contrast, services categorized as "product-oriented services" encompass activities like growth monitoring, which includes weighing children, and the distribution of supplementary food to pregnant and lactating women. These services are aimed at promoting the use of specific products through the provision of tangible goods.

## Methods

**Data.** This study used secondary data from an endline survey conducted from December 2018 through August 2019 across 12 districts of two states of northern India—Madhya Pradesh (MP) and Bihar —to evaluate the effectiveness of an mHealth intervention to digitally enable AWWs. More information is available in the published protocol [28] and the final impact evaluation paper [29]. The analytical sample consisted of 6653 mothers of children below 12 months, 2398 pregnant women, and 1344 AWW from 841 villages across 12 districts (six of which had implemented the intervention and six districts that had not implemented it ). In each village, the list of beneficiaries available with the AWW was used as a unit-level sample frame to randomly sample up to three pregnant women in the third trimester and eight mothers of children <12 months. All study participants provided verbal audio-recorded informed consent before data collection. Trained female enumerators surveyed respondents using structured computer-assisted personal interviews. Through

the research, the study team only had access to de-identified data which was available to us because our team was responsible for primary data collection.

We used the complete endline survey covering both intervention and comparison districts and controlled for the effect of the intervention using district fixed effects as the intervention was delivered at the district-level. We merged the house-hold and AWW datasets to generate a household-level dataset containing information on both beneficiaries and their AWWs. At endline, we surveyed 1344 AWW and a total of 9033 women, including 6,653 mothers of children < 12 months and 2,398 pregnant women.

**Ethical approvals.** Study protocols were reviewed and approved by institutional review boards at the University of California, Berkeley (Ref. No. 2016-08-9092), and the India-based Suraksha Independent Ethics Committee (Protocol No. 2016-08-9092). The trial is registered at https://doi.org/10.1186/ISRCTN83902145.

**Independent variables.** We mapped our independent variables onto the MMO framework described earlier based on the potential factors that could be associated with AWWs' performance.

- **Means** hypothesized to influence their service delivery included:

  - *AWW education:* This indicator variable was coded 1 if AWW had 10 or more years of schooling

  - *AWW years of experience:* This variable captures years of experience in years.

  - *Knowledge of AWWs*: This variable was created using an additive score of 11 questions ranging from 0-42 created from survey questions that examined AWW knowledge related to complementary feeding, breastfeeding, newborn care, birth preparedness, and family planning.

  - *AWW technical skills*: Depending on the outcome, the growth monitoring skills score (using vignettes on calculating and plotting growth chart) and counseling skills score (based on situational vignettes to provide life-stage appropriate messages) were examined as continuous scores.

- **Motives** closely related with service delivery included:

  - *Motivation of AWW*: This variable was coded as an indicator variable using AWW's agreement to a Likert scale question that examined the extent to which AWW felt motivated to serve their community. Coded as 1 if AWW felt motivated, 0 otherwise.

  - *Timely receipt of AWW salary*: Timely receipt of monthly salary at least in the previous 11 months was coded as 1, and 0 otherwise.

  - *Supervision of AWW:* This variable was coded as an indicator variable and coded as 1 if at least one monthly visit was made by the supervisor to the Anganwadi Center, 0 otherwise.

- **Opportunity** hypothesized to influence their service delivery included:

  - *Caste of AWW* (For caste, following the Government of India classifications, we categorized scheduled tribe/scheduled caste as marginalized caste with other backward classes and general caste serving as the reference group. Scheduled Caste and Scheduled Tribe are official Government of India caste classifications for groups of historically disadvantaged people.): This variable was constructed as an indicator variable and coded 1 if the caste of the AWW belonged to Scheduled Caste/Scheduled Tribe, 0 otherwise.

  - *Training of AWWs*: This variable was created as an indicator variable coded 1 if the AWW reported receiving training on all 23 standard modules within the previous 12 months, and 0 if they received training on only some modules. These training modules covered nutritional services (e.g., growth monitoring, complementary feeding), maternal and

child health (e.g., antenatal care, breastfeeding), community health activities (e.g., organizing Village Health and Nutrition Days), and other topics including hygiene and sanitation.

- *Workload*: We created a summation of the total number of beneficiaries, including pregnant, lactating women and children 0-6 years in the AWW's catchment area, a reflection of AWW workload.

- *Facilities and supplies* available to AWW at the AWC: For services where facilities and supplies were necessary, we controlled for sufficient indoor space for children, storage space to hold take-home rations, and salter scales meant to weigh children. For instance, for growth monitoring, we specifically controlled for the presence of salter scale used to weigh children.

**Dependent variables.** We developed the AWW performance or service delivery outcomes according to services expected of AWWs per the ICDS program, using beneficiaries' recall of service receipt and their characteristics. The analytic sample for each outcome was restricted based on the eligibility to receive the services. For example, pregnant women were not included in the denominator on growth monitoring services. Detailed construction and definition of each of the indicators are provided in the Supplementary Material (S1 Table).

**Product-oriented performance outcomes:**

1. *Growth monitoring services*: Mothers of children 0–12 months who reported receiving weighing services by the AWW per the ICDS program guidelines were coded 1, 0 otherwise.

2. *Take Home Rations:* Mothers of children 0–12 months and pregnant women who reported receiving take home rations were coded as 1, 0 otherwise.

3. *Hot Cooked Meals:* Pregnant women who reported receiving hot cooked meals at the AWC were coded as 1, 0 otherwise.

**Information-oriented performance outcomes for all beneficiaries (mothers of children <12 m and pregnant women):**

4. *Adequate quantity of home visits*: Beneficiaries who received adequate quantity of visits from AWWs were coded as 1, 0 otherwise.

5. *Counseling on IYCF*: Beneficiaries who received advice/counseling on IYCF from AWWs were coded as 1, 0 otherwise.

**Covariates.** We controlled for several potentially confounding variables as informed by prior literature [5], including: woman's age at time of survey (continuous) and woman's education (continuous). Based on a prior AWW qualitative study from India, we controlled for caste of the household [25], using an indicator variable to indicate the historically marginalized scheduled caste/scheduled tribe (SC/ST) for the households belonging to SC/ST. We accounted for household's wealth index (in quartiles) by using the principal components analysis to estimate wealth quartile with information derived from household assets and characteristics based on women's surveys. Since the mHealth intervention was assigned at the district level, we included district-level fixed effects to ensure that the associations are estimated in the original treatment and comparison districts separately and then averaged over.

## Analysis

We described AWW and AWC characteristics, using proportions or percentages for categorical variables and medians/interquartile ranges for continuous variables. We also descriptively examined women's reports of receiving nutrition services from AWW based on ICDS guidelines. We fitted a series of multivariate logistic regression models for each of the

binary outcomes and estimated odds ratios and confidence intervals. For each of the outcomes, we specified a slightly different set of independent variables based on the nature of service delivery (product-oriented or information-oriented) [5]. For instance, growth monitoring could necessitate facilities and resources such as salter scales for weighing children.

All models were adjusted for age at the time of endline survey, women's education, caste, wealth index, and district fixed effects (accounting for treatment/comparison arm of the impact evaluation). We assessed the multicollinearity of the predictors by calculating variance inflation factors (VIFs). We used a VIF greater than five as the criterion and found no multicollinearity concerns. We set the level of statistical significance (Wald p-value) at 0.05 for regression models. All models accounted for the survey study design and used cluster-robust standard errors at the village level. All analyses were conducted using Stata V.15 [30]. We added a supplementary table (S2 Table) showing the percentage and number of beneficiaries receiving each service stratified by key AWW characteristics that align with our Means, Motives, and Opportunity framework.

## Results

Table 1 summarizes the key independent variables and characteristics of the 1344 AWWs. Over half the AWW were below 38 years old, with a majority (81.4%) having completed 10 or more years of schooling. In terms of caste, 38.1% were from Other Backward Class, 37.2% from Scheduled Caste/Scheduled Tribe (marginalized caste), and approximately a quarter (24.7%) from the General Caste. The median years of experience of AWW was 14 (IQR: 9–20). Fifty percent of the AWWs had a knowledge score of below 30 (out of a maximum possible score of 47) and skills score of 3 (maximum possible score of 3) in growth monitoring and 16 ( maximum possible score of 27) in infant and young child feeding practices. The majority (92%) of AWW felt motivated about their work. Less than half received their salaries regularly, and just over half received supervision. About 50% of AWW were trained in all the health and nutrition topics. In terms of workload, AWW typically had around 73 beneficiaries to serve in their villages. Other AWC characteristics used as independent variables are also provided in Table 1.

Table 2 describes women's report of receipt of product-oriented and information-oriented services from AWW. There was a wide variation in women's receipt of different services. About 47% of women reported their children were weighed, though only a little over a third AWW (35%) discussed the weight of children with mothers. Receipt of take-home rations (THR) was the highest level of service provided, with over one in two (52.2%) pregnant women and mothers reporting receiving THR. In contrast, only one in five pregnant women (20.4%) reported receiving hot cooked meals. In terms of home visits, approximately 37% of pregnant women and mothers received an adequate number of home visits from the AWWs per the program guidelines. Forty-five percent of pregnant women and mothers received counseling on topics related to nutrition and infant and young child feeding (IYCF) practices.

### Determinants of product-oriented service delivery

**Growth monitoring services.** The results of the multivariate logistic regression analysis for growth monitoring of children 0−12 months are provided in Table 3. AWW with 10 or more years of education (AOR:1.30; 95%CI: 1.04–1.61) had greater likelihood of providing growth monitoring services, controlling for other factors and women's socio-economic characteristics, caste, and age. None of the factors related to motives were significantly associated. AWW belonging to marginalized caste (SC/ST) had 14% lower odds (AOR: 0.86; 95%CI: 0.75–0.99) of performing growth monitoring services. Presence of resources such as salter scales in the AWC (AOR: 1.30; 95%CI: 1.04–1.63) and AWW training (AOR: 1.31; 95% CI: 1.11–1.54) were associated with greater odds of beneficiaries receiving growth monitoring services.

**Supplementary food services.** None of the factors conceptualized under means domain such as education, experience, knowledge, or skills were significantly associated with odds of providing THR to beneficiaries and hot cooked meals to pregnant women. Coming to motives, AWW who received supervision had 52% greater odds

**Table 1. Characteristics of AWW and AWC in the sample (N = 1344).**

| AWW/ AWC characteristics | Value |
|---|---|
| Age (median, IQR) | 38 (32-45) |
| Completed 10 or more years of schooling, n (%) | 1094 (81.4%) |
| Caste, n (%) | |
| General Caste | 311 (24.7%) |
| Other Backward Class | 511 (38.1%) |
| Scheduled Caste/Scheduled Tribe | 500 (37.2%) |
| Years of experience as an AWW (median, IQR) | 14 (9-20) |
| Knowledge score (max possible = 47) (median, IQR) | 30 (26-33) |
| Skills score on growth monitoring (max possible = 3) (median, IQR) | 3 (2-3) |
| Skills score on infant and young child feeding practices (max possible = 27) (median, IQR) | 16 (14-17) |
| Felt motivated, n (%) | 1238 (92.1%) |
| Received timely salaries, n (%) | 610 (45.4%) |
| Received monthly supervision, n (%) | 746 (55.5%) |
| Trained in all the key health and nutrition topics, n (%) | 691 (51.4%) |
| Total number of beneficiaries reflecting workload (median, IQR) | 73 (50-96) |
| Embedded within their communities and feel valued | 1256 (93.4%) |
| Facilities, resources, and supplies n (%) | |
| Faced THR supply issue in the 6 months prior to the survey | 329 (24.4%) |
| AWC with salter scale | 1173 (87.2%) |
| AWC with storage space for food supplements | 812 (60.4%) |

**Table 2. Percentage of women reporting receipt of nutrition service delivery from AWW.**

| Services received by households | | N | n (%) |
|---|---|---|---|
| *Product-oriented service* | | | |
| Growth monitoring | Children 0–12 months weighed by AWW | 6635 | 3148 (47.5%) |
| Supplementary food | Received take home rations (THR) for woman/child one or more times a month from AWC | 9033 | 4716 (52.2%) |
| | Received hot cooked meals for herself at the AWC (only pregnant women) | 2398 | 488 (20.4%) |
| *Information-oriented service* | | | |
| Home visits | Received adequate number of home visits from AWWs based on ICDS guidelines | 9033 | 3363 (37.2%) |
| Counseling on nutrition | Received counseling on nutrition-related topics, including IYCF | 9033 | 4070 (45.0%) |

Note: The text and table refer only to pregnant women because ICDS provides THR to children 6 months to 3 years. Hot cooked meals are offered to children attending the AWC (3–6 years).

(AOR: 1.52; 95%CI: 1.18–1.96) of providing hot cooked meals to pregnant women at the Center controlling for other factors and women's socio-economic characteristics, caste, and age. Under opportunity domain, training of AWW was positively associated with provision of THR (AOR:1.36; 95%CI: 1.18–1.56) and hot cooked meals (AOR:1.64; 95%CI: 1.24–2.18). Further, AWW who faced supply issues with food supplies had 14% lower odds (AOR: 0.86; 95%CI: 0.74–0.99) of providing THR to beneficiaries. However, other aspects such as AWW caste, storage space at AWC, or workload were not significantly associated with provision of THR and hot cooked meals.

**Table 3. Multivariate logistic regression model examining AWW factors and product-oriented nutrition service delivery outcomes for mothers of children 0-12 months and/or pregnant women, Bihar and Madhya Pradesh, India.**

| Population | Growth monitoring | Take home rations | Hot cooked meals |
|---|---|---|---|
| | Children 0–12 months | Pregnant women and mothers of children 0-12m | Pregnant women only |
| | OR (95% CI) | OR (95% CI) | OR (95% CI) |
| **Means** | | | |
| 10 or more years of education | 1.30** | 1.09 | 1.29 |
| | (1.04 - 1.61) | (0.89 - 1.34) | (0.88 - 1.89) |
| Experience (in years) | 1.01 | 1.00 | 1.00 |
| | (1.00 - 1.02) | (0.99 - 1.01) | (0.98 - 1.01) |
| Knowledge score | 1.01 | 1.00 | 1.00 |
| | (0.99 - 1.02) | (0.99 - 1.02) | (0.97 - 1.03) |
| Skills score | 1.05 | 1.01 | 0.99 |
| | (0.97 - 1.14) | (0.99 - 1.03) | (0.96 - 1.02) |
| **Motives** | | | |
| Motivation | 0.87 | 0.97 | 1.12 |
| | (0.66 - 1.15) | (0.78 - 1.22) | (0.65 - 1.95) |
| Timely salary | 1.07 | 1.00 | 1.06 |
| | (0.88 - 1.30) | (0.83 - 1.20) | (0.76 - 1.48) |
| Supervision received | 1.01 | 1.14* | 1.52*** |
| | (0.86 - 1.18) | (0.99 - 1.31) | (1.18 - 1.96) |
| **Opportunities** | | | |
| SC/ST | 0.86** | 1.06 | 0.95 |
| | (0.75 - 0.99) | (0.94 - 1.20) | (0.73 - 1.24) |
| AWC has salter scale for weighing | 1.30** | | |
| | (1.04 - 1.63) | | |
| AWC has THR storage space | | 0.90 | 0.95 |
| | | (0.78 - 1.03) | (0.73 - 1.23) |
| AWW has THR supply issues | | 0.86** | 1.23 |
| | | (0.74 - 0.99) | (0.91 - 1.65) |
| Training | 1.31*** | 1.36*** | 1.64*** |
| | (1.11 - 1.54) | (1.18 - 1.56) | (1.24 - 2.18) |
| Workload | 1.00* | 1.00* | 1.00 |
| | (0.99 - 1.00) | (0.99 - 1.00) | (0.99 - 1.00) |
| Fixed effects for district | YES | YES | YES |
| Observations (N) | 6635 | 9,033 | 2,398 |

\*\*\* p < 0.01, \*\* p < 0.05, \* p < 0.1 and robust confidence interval in parentheses

AWW = Anganwadi Workers; AWC = Anganwadi Center.

Controlled for household wealth index, women's age, women's education, and household's caste.

## Determinants of information-oriented service delivery

The determinants of information-oriented service delivery are provided in Table 4.

*Home visits*. Supervision of AWW was associated with 15% greater odds of conducting an adequate number of home visits (AOR: 1.15; 95%CI: 1.02–1.30), controlling for other factors and women's socio-economic characteristics, caste, and age. However, none of the other factors conceptualized under means such as education, experience, knowledge, or skills

**Table 4. Multivariate logistic regression model examining AWW factors and information-oriented nutrition service delivery outcomes for mothers of children 0-12 months and pregnant women, Bihar and Madhya Pradesh, India.**

| | Adequate number of home visits | Counseling on IYCF |
|---|---|---|
| **Population** | Pregnant women and Mothers of children 0–12 months | |
| | OR (95% CI) | OR (95% CI) |
| **AWW Means** | | |
| 10 or more years of education | 1.04 | 1.00 |
| | (0.88 - 1.22) | (0.84 - 1.20) |
| Experience (in years) | 1.00 | 1.01 |
| | (0.99 - 1.01) | (1.00 - 1.01) |
| Knowledge score | 1.00 | 1.01 |
| | (0.99 - 1.02) | (0.99 - 1.02) |
| Skills score | 1.00 | 1.01 |
| | (0.99 - 1.02) | (0.99 - 1.02) |
| **AWW Motives** | | |
| Motivation | 1.03 | 1.30** |
| | (0.82 - 1.28) | (1.05 - 1.61) |
| Timely salary | 1.13 | 1.19** |
| | (0.95 - 1.33) | (1.01 - 1.41) |
| Supervision received | 1.15** | 1.14** |
| | (1.02 - 1.30) | (1.01 - 1.30) |
| **AWW Opportunities** | | |
| SC/ST | 0.89** | 0.98 |
| | (0.79 - 1.00) | (0.88 - 1.11) |
| Training | 1.08 | 1.31*** |
| | (0.95 - 1.22) | (1.15 - 1.49) |
| Workload | 1.00 | 1.00 |
| | (1.00 - 1.00) | (1.00 - 1.00) |
| Fixed effects for district | YES | YES |
| Observations | 9,033 | 9,033 |

*** $p < 0.01$, ** $p < 0.05$, * $p < 0.1$ and robust confidence interval in parentheses.

AWW = Anganwadi Worker; AWC = Anganwadi Center.

Controlled for household wealth index, women's age, women's education, and household's caste.

were associated with adequate number of home visits. Under opportunity, the only significant predictor was AWW caste— AWW belonging to the marginalized caste (SC/ST) had 11% lower odds of providing an adequate number of home visits (AOR: 0.89; 95%CI: 0.79–1.00).

**Counseling on IYCF.** When examining IYCF counseling, none of the AWW capability factors (education, experience, knowledge, or skills) were significantly associated with service delivery. Controlling for women's socio-economic characteristics and age, all the factors conceptualized under motives domain were significantly associated with higher provision of counseling on IYCF—higher motivation (AOR: 1.30; 95%CI: 1.05–1.61), timely receipt of salaries (AOR: 1.19; 95%CI: 1.01–1.41), and supervision (AOR: 1.14; 95%CI: 1.01–1.30). Further, under opportunity, we found that training of AWW(AOR:1.31; 95%CI: 1.15–1.49) was positively associated with receipt of counseling on IYCF. But none of the other factors such as workload were associated with women's receipt of counseling on IYCF.

## Discussion

This paper investigated the determinants of CHW performance in one of the largest CHW programs globally. By adapting the Means, Motives, and Opportunity framework, we examined the factors associated with AWW performance on five important ICDS services, including product-oriented and information-oriented services using maternal receipt of services self-reported by beneficiaries. Overall, the results highlighted that provision and coverage of ICDS services remain low. We found that opportunity-related factors—such as AWW caste, training, and available facilities and resources—were more significantly associated with AWW performance for product-oriented services than factors categorized under means and motives domains. Further, motives and opportunity-related factors were important predictors of information-oriented services such as home visits and counseling.

Our findings on ICDS service coverage show some alignment with patterns observed in NFHS-5 data, though with notable variations. In Madhya Pradesh and Bihar, NFHS-5 reported that 75.9% and 41.3% of children under 6 years, respectively, received supplementary food through ICDS, averaging to 58.6% across both states – somewhat higher than our observed THR receipt rates (52.2%). For pregnant women, NFHS-5 found that 43.8% in Bihar and 83.8% in MP received supplementary food during pregnancy, substantially higher than the hot cooked meals (20.4%) reported in our study. This difference likely reflects that NFHS-5 defines supplementary food to include both food cooked and served at the AWC and take-home rations. Regarding growth monitoring, NFHS-5 showed that 35.2% in Bihar and 77.8% in MP of children were weighed by AWW, compared to 47.5% in our sample, placing our findings between the two states' averages. NFHS-5 does not provide data on home visits or nutrition counseling, preventing comparison for these services. These patterns suggest regional variations in ICDS implementation, with our findings generally reflecting the broader service delivery landscape across these states [17].

### Means

Of the five outcomes across both product and information-oriented services, only education was significantly positively correlated with higher provision of growth monitoring services, but none of the other AWW characteristics such as experience, knowledge, or skills were associated with the remaining services. The association between AWW education and growth monitoring services specifically may reflect the technical nature of this task, which requires accurate measurement, interpretation of growth charts, and record-keeping – skills that might be enhanced through formal education. This finding suggests that educational criteria in AWW recruitment may be particularly important for services requiring technical competencies, though our results indicate education alone is insufficient to ensure comprehensive service delivery across all ICDS program components. This also suggests that while the matriculation (completing tenth grade) requirement may be beneficial for certain technical aspects of the AWW role, it may not be uniformly necessary across all responsibilities. This finding has important implications for AWW recruitment policies, particularly in areas with limited candidate pools, and suggests that task-specific training and supportive supervision may potentially compensate for lower formal education in some aspects of service delivery. One previous systematic review on CHW found that more years of education is associated with higher performance of CHW while research on experience of CHW remains mixed [3]. We did not observe any association between knowledge and skills and any of the outcomes. In other studies [19,31], including one from Bangladesh, authors highlighted differences in how knowledge and skill sets were associated with performance (measured as retention) of CHW. While knowledge gain and perceived skillsets were associated with better retention of CHW in rural areas, they were not important predictors in urban settings [32].

### Motives

Among the five outcomes explored in this study, we observed that higher levels of motivation were correlated with improved provision of counseling services, but we did not observe a statistical association between motivation and the

other four outcomes studied. One limitation of our study was that we did not use a validated scale to measure motivation (which was self-reported by AWW as a response to a 5-point Likert scale), potentially resulting in a failure to accurately capture motivation of AWW. While previous studies acknowledge the significance of motivation in CHW performance [3,21,33], there is also an acknowledgement that motivation is a complex, multifaceted topic that may be impacted by individual-level and organizational factors [33,34]. For instance, an Indian study found that AWW motivation may be influenced by their interpersonal factors such as family support and other skill development opportunities [34].

Supportive supervision was found to be significantly linked with multiple performance metrics including increased provision of hot cooked meals, conducting home visits, and counseling mothers on IYCF. Prior studies from India corroborate that supportive supervision was significantly associated with improved CHW service delivery, mediated by increased knowledge of CHWs [9,10,35]. Timely receipt of salaries by the AWW, an indicator of financial motive, was associated with higher provision of counseling services. Our results were consistent with a prior study from India that also noted financial incentives as being an important predictor of households receiving general nutrition information [5]. A qualitative study from Bihar asserted that financial motives, especially as a means of livelihood to support the family, were an important driver of AWW performance. However, we did not observe a significant association between timely receipt of salaries and provision of the other four outcomes. A few global reviews noted that the evidence on financial incentives for determining CHW performance was mixed [3,5,31]

## Opportunity

Overall, training emerged as a significant factor associated with four of the five services, including provision of growth monitoring services, THR, hot cooked meals, and counseling services. This corroborates prior evidence reviews that have identified continuous and ongoing training as an influential factor for CHW performance due to the evolving nature of service delivery and associated guidelines [36,37]. Further, lack of facilities and insufficient supplies were negatively associated with growth monitoring and provision of THR, as would be expected. Potentially, lack of facilities could have also demotivated AWW from providing some of the services.

Interestingly, our findings suggest that AWW belonging to Scheduled Caste/Scheduled Tribe, the most marginalized castes, were negatively associated with two services—provision of growth monitoring (weighing of children) and counseling of beneficiaries, even after controlling for household caste. This observed association may reflect complex social dynamics that quantitative analysis alone cannot fully explain. While previous qualitative research from Bihar suggests that caste-based social hierarchies can affect service delivery [25], other unmeasured factors correlated with AWW caste may also contribute to these differences. Future research using mixed methods would be valuable to better understand the specific mechanisms through which AWW caste relates to service delivery, particularly examining how institutional, community, and interpersonal factors interact to create barriers or facilitators for AWW from different caste backgrounds. Unsurprisingly, prior research from India has noted that caste of the AWW and household were important determinants of service delivery [5,25,26], particularly home visits [26]. For instance, a qualitative study conducted in Bihar, one of our study states, found that higher-caste communities may not always accept services provided by lower-caste AWW. Additionally, they observed that even AWW from higher-caste backgrounds may struggle to meet the expectations of both lower-caste and their own caste group members [25]. Another study noted that frequency of visits by the AWW was greater in higher caste households, although ICDS service utilization was better for lower caste households [38]. Caste-based sensitization and training of AWW could be considered to avoid personal biases at the level of AWW, though changing societal prejudice may take time.

## Strengths and limitations

Our study had some limitations, including potential recall bias and social desirability bias due to reliance on self-reported data from AWW and mothers. For instance, AWW self-reported motivation likely reflected a degree of social desirability

bias. We attempted to mitigate this by using trained female enumerators and privacy during interviews, but recognize this remains a limitation, particularly for subjective measures. As this study was embedded within a larger evaluation of an mHealth intervention targeting AWW, it's possible that both AWW and beneficiaries may have altered their behavior due to awareness of being observed. We attempted to control for this by including district fixed effects in our models, as the intervention was assigned at the district level. However, we cannot completely rule out potential reactivity effects. It is also important to note that our findings, while based on a substantial sample of over 6,500 mothers and 2,500 pregnant women from 841 villages, were not strictly representative of the entire population of mothers with children under 12 months in Bihar and MP. This is because our sample was based on propensity score matching and included individuals exposed to an mHealth intervention.

Regarding the time gap between data collection (2018–19) and analysis, it is important to note that this temporal delay did not affect the internal validity of the associations we report. The data represents conditions at the time of collection, and the relationships between AWW characteristics and service delivery outcomes remained valid regardless of when the analysis was performed. An additional limitation is that we did not examine the full spectrum of ICDS services. Our data collection instruments were developed to capture services primarily delivered to mothers rather than direct child feeding programs at AWC. We focused only on nutrition-related services targeting the first 1000 days of life (growth monitoring, supplementary food, home visits, and IYCF counseling), which aligned with our study population of mothers with children under 12 months and pregnant women. Pre-school education services and cooked meals for children at AWC would have required data from families with children aged 3–6 years. Services like immunization (delivered collaboratively with ASHAs and ANMs) and referral services (which represent outcomes rather than distinct service actions) were outside our scope. Our results are not generalizable to the full range of services offered by the AWW. Future studies should examine the full range of ICDS services.

We used the full dataset of both intervention and control districts to retain the full sample and addressed this by controlling for the district fixed effects as the intervention was delivered at the district level. This statistical approach effectively controlled for any potential influences of the mHealth intervention, which was delivered at the district level. By incorporating these fixed effects, we essentially compared AWW within the same sub-districts, thereby accounting for both intervention-related factors and sub-district level characteristics that might influence service delivery patterns or women's health-seeking behaviors. Further, by using beneficiary-reported outcomes rather than program administrative data, our measures capture actual service receipt from the perspective of intended beneficiaries, which provides a more realistic assessment of service delivery irrespective of the intervention. This approach helps mitigate potential reporting biases that might arise from AWW themselves in an intervention context.

Finally, we drew upon a large and uniquely structured dataset that had a wide range of factors to study AWW core nutrition service delivery, as defined in the ICDS program. Therefore, the factors highlighted as important in this study may offer guidance on how to maximize investments in the ICDS programs and AWW performance improvement. Even though we have many factors, we still did not have data to control for gender norms and other contextual factors that had been noted in the literature as potentially influencing CHW performance.

## Conclusion

CHW are gaining attention in the context of national and global objectives, including Sustainable Development Goals, Universal Health Coverage, and ending preventable child and maternal deaths [39]. Leveraging CHW capabilities and performance can overcome health system challenges and expand program outreach [40], expediting advancements toward achieving Universal Health Coverage (UHC). Our findings suggest that opportunity-related factors—particularly training and availability of facilities and supplies—are critical determinants of AWW performance. We also identified AWW characteristics such as caste and education level as important predictors of service delivery. These results highlight the need for a multi-faceted approach to improving ICDS program implementation that addresses both systemic factors (ensuring

adequate training, supervision, and infrastructure) and structural barriers related to social hierarchies. The association between AWW education and service delivery, particularly for technically demanding tasks, reinforces the broader importance of women's education in strengthening community health systems. Policymakers should consider these findings when designing strategies to enhance AWW performance and maximize the impact of one of the world's largest community health worker programs.

## Supporting information

**S1 Table. Detailed construction of outcome indicators and definitions.**
(DOCX)

**S2 Table. Supplementary Table showing frequencies and percentages of services received by beneficiaries based on AWW's means, motives, and opportunity variables.**
(DOCX)

## Acknowledgments

We extend our sincere thanks to the enumerators at NEERMAN for data collection. Most of all, we thank the survey respondents for their time. We also acknowledge contributions made by colleagues at International Food Policy and Research Institute, NEERMAN, University of California San Francisco, and University of California at Berkeley. We also thank Sneha Nimmagadda at NEERMAN for her efforts in cleaning and preparing the dataset.

## Author contributions

**Conceptualization:** Lakshmi Gopalakrishnan.

**Formal analysis:** Lakshmi Gopalakrishnan.

**Funding acquisition:** Lia Fernald, Dilys Walker.

**Methodology:** Lakshmi Gopalakrishnan.

**Supervision:** Sumeet Patil, Nadia Diamond-Smith.

**Writing – original draft:** Lakshmi Gopalakrishnan.

**Writing – review & editing:** Lakshmi Gopalakrishnan, Sumeet Patil, Lia Fernald, Dilys Walker, Nadia Diamond-Smith.

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
