## [Decision Letter · Decision Letter 0]

Dear Dr. Gopalakrishnan,

Thank you for submitting your manuscript to PLOS ONE. After careful consideration, we feel that it has merit but does not fully meet PLOS ONE’s publication criteria as it currently stands. Therefore, we invite you to submit a revised version of the manuscript that addresses the points raised during the review process.

We look forward to receiving your revised manuscript.

Kind regards,

Rakesh Sarwal, MBBS, MPH, DrPH, ThYC

Academic Editor

PLOS ONE

Journal Requirements:

3. Please include captions for your Supporting Information files at the end of your manuscript, and update any in-text citations to match accordingly. Please see our Supporting Information guidelines for more information: http://journals.plos.org/plosone/s/supporting-information .

Additional Editor Comments:

The article covers a very important topic of immediate relevance. It could be made even better by addressing following four issues:

1. Mode of recruitment of AWW decide on their capability, and also play a huge influence on their motivation. Can a para be added on government policy and practice on how AWW are recruited, whether Recruitment Rules are framed and followed.

2. Compare data on services with that from NFHS-5.

3. Caste composition of served population is relevant when evaluating performance of AWW.

Reviewers' comments:

Reviewer's Responses to Questions

**Comments to the Author**

1. Is the manuscript technically sound, and do the data support the conclusions?

Reviewer #1: Yes

2. Has the statistical analysis been performed appropriately and rigorously?

Reviewer #1: Yes

3. Have the authors made all data underlying the findings in their manuscript fully available?

Reviewer #1: Yes

4. Is the manuscript presented in an intelligible fashion and written in standard English?

Reviewer #1: Yes

Reviewer #1: The authors have chosen a very relevant topic and have meticulously conducted extensive background work on the same. They have identified various domains of determinants and outcomes for addressing this research question. This is a descriptive study with an analytical component to look for factors associated with the service delivery of Integrated Child Development Services (ICDS). Secondary data analysis has been done. There has been no data collection specifically to answer this research question. The paper has been written very well.

A few queries

1. The Caste variable – it has been speculated in the article that the reason AWWs from SC/ST backgrounds are unable to perform their responsibilities concerning growth monitoring and home visits is because of caste dynamics. However, it has also been mentioned in the paper that the model was adjusted for Caste status. It is a very sensitive topic and needs to be treated carefully.

2. The paper has identified that ‘training’ is an essential determinant of the service delivery of ICDS. However, the methods section does not elaborate on what this training entails. Is it ever trained/never trained? Or Trained in the last 1-year vs more than a year ago? Also the contents of the training, taught by whom etc. This is essential as it has become the single most influential factor across domains.

3. Would it be reasonable to say that the study concludes that for an AWW to function well, they must be educated for more than 10 years? Therefore, can that be a criterion to consider when choosing AWWs? It is a thought for the authors to ponder.

4. As mentioned in the paper, information bias is a significant worry in this particular study design. The paper has also been completed 5 years after the original study concluded. This may add to the bias. The question on ‘motivation’ is bound to have a substantial social desirability bias. In addition, there is a possibility of Hawthorne bias as this was a part of another study on the usage of a mobile app.

5. It would have been more illuminating to have the actual frequencies and percentages in addition to the Odds Ratios.

**Do you want your identity to be public for this peer review?** For information about this choice, including consent withdrawal, please see our Privacy Policy

Reviewer #1: **Yes: ** Jackwin Sam Paul G

---

## [Author Response · Author response to Decision Letter 1]

5 Apr 2025

Dear Reviewer and Editor:

We appreciate the thorough and constructive feedback provided by the reviewer and the editor. We are particularly encouraged by reviewer’s positive assessment that “The authors have chosen a very relevant topic and have meticulously conducted extensive background work on the same.” Further, the editor too noted: “The article covers a very important topic of immediate relevance”. We have carefully considered all comments and suggestions and have made appropriate revisions to strengthen the manuscript. Our responses are in red below. Page numbers where we made edits are highlighted in yellow in the track change version to easily find the edits.

Reviewer #1: The authors have chosen a very relevant topic and have meticulously conducted extensive background work on the same. They have identified various domains of determinants and outcomes for addressing this research question. This is a descriptive study with an analytical component to look for factors associated with the service delivery of Integrated Child Development Services (ICDS). Secondary data analysis has been done. There has been no data collection specifically to answer this research question. The paper has been written very well.

A few queries

1. The Caste variable – it has been speculated in the article that the reason AWWs from SC/ST backgrounds are unable to perform their responsibilities concerning growth monitoring and home visits is because of caste dynamics. However, it has also been mentioned in the paper that the model was adjusted for Caste status. It is a very sensitive topic and needs to be treated carefully.

Response to reviewer: Thank you for this insight. Yes, you’re right that we controlled for household caste and still found that AWW caste was associated with service delivery outcomes, so we have now revised our discussion of this finding to more carefully present the possible interpretations.

First, we acknowledge the fact that we controlled for household caste. Our findings indicate that AWWs from Scheduled Caste/Scheduled Tribe backgrounds had lower odds of providing growth monitoring services and conducting home visits, even after controlling for household caste. We further added:

“This observed association may reflect complex social dynamics that quantitative analysis alone cannot fully explain. While previous qualitative research from Bihar (John et al., 2020) suggests that caste-based social hierarchies can affect service delivery, other unmeasured factors correlated with AWW caste may also contribute to these differences. Future research using mixed methods would be valuable to better understand the specific mechanisms through which AWW caste relates to service delivery, particularly examining how institutional, community, and interpersonal factors interact to create barriers or facilitators for AWWs from different caste backgrounds.”

Please refer to Page 14 of 19 for revised discussion paragraph on caste dynamics.

2. The paper has identified that ‘training’ is an essential determinant of the service delivery of ICDS. However, the methods section does not elaborate on what this training entails. Is it ever trained/never trained? Or Trained in the last 1-year vs more than a year ago? Also the contents of the training, taught by whom etc. This is essential as it has become the single most influential factor across domains.

Response to reviewer: Thank you for highlighting this important point. In the revised manuscript, we have included this detailed explanation in the methods section to provide readers with a clearer understanding of how this key variable was operationalized, especially given its significant association with service delivery outcomes across multiple domains. Page 6 of 18 clearly defines how training variable was coded.

To clarify, our training variable was defined as an indicator variable coded as 1 if the AWW reported receiving training on all 23 modules within the 12 months prior to the survey, and 0 if they received training on some but not all modules. This binary categorization ("Trained in all" vs. "Trained in some") allowed us to assess the impact of comprehensive training coverage. The 23 training modules delivered by supervisors covered a broad range of AWW responsibilities including:

1. Nutritional services: Growth monitoring, complementary feeding, management of malnutrition (SAM), feeding during illness, maternal nutrition during pregnancy and lactation

2. Maternal and child health: Antenatal care, postnatal care, breastfeeding, immunization, family planning, identification of sick newborns

3. Community health activities: Organizing Village Health and Nutrition Days (VHNDs), home visit scheduling, referrals to health centers

4. Other topics: Hygiene, sanitation, adolescent health

Our approach focused on comprehensiveness of training rather than recency, as all training modules were delivered within the previous 12 months.

3. Would it be reasonable to say that the study concludes that for an AWW to function well, they must be educated for more than 10 years? Therefore, can that be a criterion to consider when choosing AWWs? It is a thought for the authors to ponder.

Response to reviewer: Thank you for this suggestion. Please refer to Page 13 of 18 under Motives paragraph.

Our findings do indicate that AWWs with 10 or more years of education had 30% higher odds of providing growth monitoring services compared to those with less education. This aligns with some previous literature suggesting education may enhance CHW performance. However, based on these results alone, it would not be possible for us to recommend education as a strict criterion for reasons such as:

1) Education was significantly associated only with growth monitoring services, not with other key services like supplementary food provision, home visits, or counseling. This suggests the relationship between AWW education and performance are service-specific rather than universal.

2) Further, we feel that raising the minimum educational requirements could inadvertently exclude capable candidates from communities with limited educational access for women, potentially reducing community representation among AWWs. Such requirements may pose challenges in recruitment and retention of AWWs.

3) Finally, our findings indicate that opportunity-related factors (training, supervision, adequate supplies) were more consistently associated with AWW performance across multiple services than educational background. These modifiable program factors may offer more actionable paths to improving service delivery.

Addressed this by adding a point on the importance of education but also cautioning the readers about relying solely on education as criteria for performance.

4. As mentioned in the paper, information bias is a significant worry in this particular study design. The paper has also been completed 5 years after the original study concluded. This may add to the bias. The question on ‘motivation’ is bound to have a substantial social desirability bias. In addition, there is a possibility of Hawthorne bias as this was a part of another study on the usage of a mobile app.

Response to reviewer:

Thank you for highlighting these important methodological considerations. We acknowledge several potential sources of bias in our study:

1. Information bias: We recognize the limitations of self-reported data from both AWWs and beneficiaries. To minimize recall bias, we focused on recent service delivery experiences and incorporated data from multiple stakeholders (mothers, pregnant women, and AWWs).

2. Social desirability bias: We acknowledge that measures such as AWW motivation likely reflect some degree of social desirability bias. We attempted to mitigate this by using trained female enumerators and privacy during interviews, but recognize this remains a limitation, particularly for subjective measures.

3. Timing of analysis: While the data collection was completed in 2018-19 and the analysis conducted later, this temporal gap does not affect the internal validity of the associations we report. The data itself represents conditions at the time of collection, and the relationships between AWW characteristics and service delivery outcomes remain valid regardless of when the analysis was performed.

4. Hawthorne effect: As this study was embedded within a larger evaluation of an mHealth intervention, but the mHealth intervention was with the AWWs. However, it is possible AWWs and even to an extent, participants may have altered their behavior due to awareness of being observed. We attempted to control for this by including district fixed effects in our models, as the intervention was assigned at the district level. However, we cannot completely rule out potential reactivity effects.

Despite these limitations, the consistency of our findings with qualitative research from similar contexts and the robust associations observed across multiple outcome measures suggest that our key findings regarding training, supervision, and resource availability as determinants of AWW performance remain valid and informative for program implementation. We have addressed this in the limitations section of the paper (Page 14 of 19) in the track changes version.

5. It would have been more illuminating to have the actual frequencies and percentages in addition to the Odds Ratios.

We have now added a supplementary table (Supplementary Table S2) showing the percentage and number of beneficiaries receiving each service stratified by key AWW characteristics that align with our Means, Motives, and Opportunity framework. This descriptive analysis complements our regression results.

Additional Editor Comments:

Thank you for your thoughtful comments on our article. We have carefully addressed each point to strengthen the paper:

1. Mode of recruitment of AWW decide on their capability, and also play a huge influence on their motivation. Can a para be added on government policy and practice on how AWW are recruited, whether Recruitment Rules are framed and followed.

We have added a detailed paragraph on AWW recruitment policies in the introduction (Page 2 of 18, Paragraph 2): “AWWs are selected from local villages by State-appointed committees, with minimum qualification requirements of Matriculation (10th standard) and age limits of 18-35 years. As honorary workers rather than formal employees, AWWs receive modest honoraria (₹4,500 monthly plus ₹500 performance-linked incentive as of 2021), which may be supplemented by state governments. This recruitment approach directly influences both AWW capabilities and motivation - local recruitment enhances community embeddedness but potentially limits the educational qualifications of candidates, while the honorary worker status and compensation structure may affect motivation and retention, particularly given the substantial responsibilities assigned to AWWs (Press Information Bureau, 2021).”

2. Compare data on services with that from NFHS-5.

We have incorporated a comprehensive comparison with NFHS-5 data in our discussion section (Page 12 of 18, Paragraph 2). This comparison contextualizes our findings within national survey data: “Our findings on ICDS service coverage show some alignment with patterns observed in NFHS-5 data, though with notable variations. In Madhya Pradesh and Bihar, NFHS-5 reported that 75.9% and 41.3% of children under 6 years, respectively, received supplementary food through ICDS, averaging to 58.6% across both states - somewhat higher than our observed THR receipt rates (52.2%). For pregnant women, NFHS-5 found that 43.8% in Bihar and 83.8% in MP received supplementary food during pregnancy, substantially higher than the hot cooked meals (20.4%) reported in our study. This difference likely reflects that NFHS-5 defines supplementary food to include both food cooked and served at the AWC and take-home rations. Regarding growth monitoring, NFHS-5 showed that 35.2% in Bihar and 77.8% in MP of children were weighed by AWWs, compared to 47.5% in our sample, placing our findings between the two states’ averages. NFHS-5 does not provide data on home visits or nutrition counseling, preventing comparison for these services. These patterns suggest regional variations in ICDS implementation, with our findings generally reflecting the broader service delivery landscape across these states (International Institute for Population Sciences, 2022).”

3. Caste composition of served population is relevant when evaluating performance of AWW.

Our methodology explicitly accounts for the caste composition of the served population. As noted in our methods section (Page 7 of 18, Paragraph on Covariates), we controlled for household caste in all regression models, using an indicator variable for historically marginalized scheduled caste/scheduled tribe (SC/ST) households. Additionally, our analysis of AWW caste factors take into account this control for beneficiary caste, allowing us to identify associations between AWW characteristics and service delivery independent of the population served. This methodological approach addresses the important social dynamics you highlighted.

We believe these additions have significantly strengthened the paper while maintaining its focus on the factors associated with AWW service delivery. Thank you for your valuable feedback that has enhanced the relevance and strengthened our paper.

---

## [Editor Report · Decision Letter 1]

Dear Dr. Gopalakrishnan,

Thank you for submitting your manuscript to PLOS ONE. After careful consideration, we feel that it has merit but does not fully meet PLOS ONE’s publication criteria as it currently stands. Therefore, we invite you to submit a revised version of the manuscript that addresses the points raised during the review process.

We look forward to receiving your revised manuscript.

Kind regards,

Rakesh Sarwal, MBBS, MPH, DrPH, ThYC

Academic Editor

PLOS ONE

Journal Requirements:

Additional Editor Comments:

Appreciate the constructive manner of responding to reviewers comments, and making amends in the manuscript. The article as it exists stands adds to existing knowledge on this crucial subject. There are some suggestions (listed below) for the consideration of the authors for further improvement of the piece.

1. Intro: "Each AWW caters to a catchment area of approximately 800–1000 children under the age of six and their pregnant or breastfeeding mothers." Please recheck this fact, as program manual mentions of a POPULATION of 1000 for each cluster under the AWW. This also matches with the statement in results that "AWWs typically had around 73 beneficiaries to serve in their villages"

2. Introduction: "delivering the ICDS services: growth monitoring activities of children as appropriate; delivering supplementary food including hot cooked meals and take-home rations; conducting home visits and counseling to educate pregnant and lactating women on pregnancy care, and infant and young child feeding practices". Best to cite the standard six ICDS services here, in Figure -1, and elsewhere as these are the core ICDS services that AWW are expected to deliver.,

3. Similarly, the six outcome variables of "Product-oriented performance outcomes in the study" miss the following of the six services under ICDS

pre-school education, health check-ups, immunization, referral services.

To make the findings complete and generalizatble, some explanation on why there services were left out is required.

4. Since the data was collected at endline after a mHealth intervention, how generalizable can it to be to the conditions across ICDS project areas in general ?

5. Since around 19% of AWW were not matriculate, authors may check if this was indeed a necessary qualification.

---

## [Author Response · Author response to Decision Letter 2]

30 May 2025

Dear Editor:

We appreciate the thorough and constructive feedback provided by the editor. We have carefully considered all comments and suggestions and have made appropriate revisions to strengthen the manuscript. Our responses are in red below along with page numbers. Edits are highlighted in yellow in the track change version.

Additional Editor Comments:

Appreciate the constructive manner of responding to reviewers comments, and making amends in the manuscript. The article as it exists stands adds to existing knowledge on this crucial subject. There are some suggestions (listed below) for the consideration of the authors for further improvement of the piece.

1. Intro: "Each AWW caters to a catchment area of approximately 800–1000 children under the age of six and their pregnant or breastfeeding mothers." Please recheck this fact, as program manual mentions of a POPULATION of 1000 for each cluster under the AWW. This also matches with the statement in results that "AWWs typically had around 73 beneficiaries to serve in their villages"

Edited this to 1000 on Page 3 of 18.

2. Introduction: "delivering the ICDS services: growth monitoring activities of children as appropriate; delivering supplementary food including hot cooked meals and take-home rations; conducting home visits and counseling to educate pregnant and lactating women on pregnancy care, and infant and young child feeding practices". Best to cite the standard six ICDS services here, in Figure -1, and elsewhere as these are the core ICDS services that AWW are expected to deliver.,

Noted and cited.

3. Similarly, the six outcome variables of "Product-oriented performance outcomes in the study" miss the following of the six services under ICDS pre-school education, health check-ups, immunization, referral services.

To make the findings complete and generalizable, some explanation on why there services were left out is required.

We agree with you that not all the ICDS services are covered. Our study deliberately focused on specific nutrition-related services provided by Anganwadi Workers for several methodologically sound reasons:

1) Our study was focused on services critical to addressing malnutrition in the first 1000 days of life - a period widely recognized as the most crucial window for nutritional intervention. Growth monitoring, supplementary food distribution, home visits, and IYCF counseling represent the core nutrition-focused services that have the strongest evidence base for impact on child growth and development outcomes during this critical period. This was also the overlap with the CAS Application (mHealth intervention) designed for AWWs, and within the evaluation, we asked other questions that helped us examine research questions in this paper. Therefore, our analytical sample comprised 6,653 mothers of children below 12 months and 2,398 pregnant women. This population was specifically selected to examine services targeting pregnancy and early infancy, which are primarily nutrition-focused. Pre-school education services would require data from families with children aged 3-6 years, which was outside the scope of our targeted population.

2) While AWWs are involved in multiple ICDS services as outlined in the guidelines, not all services fall exclusively under their direct responsibility. For example:

a. Immunization services are typically delivered through a collaborative effort involving ASHAs and ANMs during Village Health and Nutrition Days, with AWWs playing a supportive rather than primary role

b. Referral services represent an outcome of other service interactions rather than a distinct service delivery action. Growth monitoring is one aspect of referral services. Based on the category of malnutrition, children are then referred by AWW, though she does not have the sole authority to ensure children end up in nutrition rehabilitation centers.

3) We were conducting the secondary data analysis from an endline evaluation of an mHealth intervention specifically designed to digitally enable AWWs in delivering nutrition services. The original data collection instruments were therefore calibrated to measure the nutrition-related services that the intervention was intended to enhance.

I have added this to the limitation on page 14 of 18 : An additional limitation is that we did not examine the full spectrum of ICDS services. We focused specifically on nutrition-related services targeting the first 1000 days of life (growth monitoring, supplementary food, home visits, and IYCF counseling), which aligned with our study population of mothers with children under 12 months and pregnant women. Pre-school education services would have required data from families with children aged 3-6 years. Services like immunization (delivered collaboratively with ASHAs and ANMs) and referral services (which represent outcomes rather than distinct service actions) were outside our scope. Our results are not generalizable to the full range of services offered by the AWWs.

4. Since the data was collected at endline after a mHealth intervention, how generalizable can it to be to the conditions across ICDS project areas in general ?

While acknowledging that the data was collected in a post-intervention context, we had the following methodological strengths that support the generalizability of our findings to the broader ICDS system.

Added this limitation on page 14 of 18: We used the full dataset of both intervention and control districts to retain the full sample and addressed this by controlling for the district fixed effects as the intervention was delivered at the district level.This statistical approach effectively controlled for any potential influences of the mHealth intervention, which was delivered at the district level. By incorporating these fixed effects, we essentially compared AWWs within the same sub-districts, thereby accounting for both intervention-related factors and sub-district level characteristics that might influence service delivery patterns or women’s health-seeking behaviors. Further, by using beneficiary-reported outcomes rather than program administrative data, our measures capture actual service receipt from the perspective of intended beneficiaries, which provides a more realistic assessment of service delivery irrespective of the intervention. This approach helps mitigate potential reporting biases that might arise from AWWs themselves in an intervention context.

5. Since around 19% of AWW were not matriculate, authors may check if this was indeed a necessary qualification.

Thank you for this insightful observation. The paper states that the minimum qualification requirement for AWWs is matriculation (10th standard), as per official ICDS guidelines. However, as correctly noted, our data shows that approximately 19% of AWWs in our sample had not completed this level of education.

This discrepancy likely reflects the practical reality of AWW recruitment in rural MP and Bihar, where policy implementation often differs from written guidelines. The ICDS program guidelines do specify matriculation as the minimum qualification, but states have historically been granted flexibility in implementation, especially in areas where finding qualified candidates willing to work for the provided honorarium has been challenging. This finding actually reinforces one of our key points about the tension between local recruitment (which enhances community embeddedness) and educational qualifications of candidates. The presence of AWWs without matriculation in our sample provides an opportunity to examine whether this official educational requirement is indeed necessary for effective service delivery. Interestingly, our analysis showed that higher education (10 or more years) was significantly associated only with growth monitoring services but not with other service outcomes. This suggests that while education may be beneficial for technically complex tasks, it may not be uniformly necessary across all AWW responsibilities, which has important implications for recruitment policies in resource-constrained settings.

I added this note in the discussion on Page 12 of 18 “This suggests that while the matriculation (completing tenth grade) requirement may be beneficial for certain technical aspects of the AWW role, it may not be uniformly necessary across all responsibilities. This finding has important implications for AWW recruitment policies, particularly in areas with limited candidate pools, and suggests that task-specific training and supportive supervision may potentially compensate for lower formal education in some aspects of service delivery.”

---

## [Editor Report · Decision Letter 2]

Dear Dr. Gopalakrishnan,

Thank you for submitting your manuscript to PLOS ONE. After careful consideration, we feel that it has merit but does not fully meet PLOS ONE’s publication criteria as it currently stands. Therefore, we invite you to submit a revised version of the manuscript that addresses the points raised during the review process.

We look forward to receiving your revised manuscript.

Kind regards,

Rakesh Sarwal, MBBS, MPH, DrPH, ThYC

Academic Editor

PLOS ONE

Journal Requirements:

Additional Editor Comments:

Appreciate the revisions and inclusions done in response to Reviewers and Editors comments,

Still, there are two issues that need authors attention:

1. Population covered by one AWW:

Its 1000 people and not 1000 children, as stated in previous comments.

2. How Cooked Meals.

The data and text covers only Pregnant women. Under ICDS guidelines, the children aged 6 months to 6 years are given Hot Cooked Meals within the AWC. The authors may like to clarify the reasons for this important omission.

---

## [Author Response · Author response to Decision Letter 3]

5 Jun 2025

Dear Editor:

We appreciate the thorough and constructive feedback provided by the editor. We have carefully considered all comments and suggestions and have made appropriate revisions to strengthen the manuscript. Our responses are in red below along with page numbers. Edits are in the track change version.

Additional Editor Comments:

1. Population served by AWW:

Its 1000 people, not 1000 children, as has erroneously appeared in your latest draft.

Apologies for missing it earlier. Edited it now.

2. Hot Cooked Meals

The text and table refers to only Pregnant Women. Why is the data for eligible children (6 months to 6 years) not included ?

The text and table refer only to pregnant women because ICDS provides THR to children 6 months to 3 years. Hot cooked meals are offered to children attending the AWC (3– 6 years). Our sample comprised 6,653 mothers of children below 12 months and 2,398 pregnant women, totaling 9,051 participants. This was because our study was specifically designed to examine services around maternal nutrition services and interventions targeting pregnancy and early infancy.

How it is address in the manuscript:

I have captured the above point in two places – one as a note to clarify that hot cooked meals are only given to children 3-6years on Page 7 of 18 in track change version.

Further in the limitations section we have mentioned our focus on the first 1000 days of life and nutrition services for women around pregnancy. “Our data collection instruments were developed to capture services primarily delivered to mothers rather than direct child feeding programs at AWC. We focused only on nutrition-related services targeting the first 1000 days of life (growth monitoring, supplementary food, home visits, and IYCF counseling), which aligned with our study population of mothers with children under 12 months and pregnant women. Pre-school education services and cooked meals for children at AWC would have required data from families with children aged 3-6 years. Services like immunization (delivered collaboratively with ASHAs and ANMs) and referral services (which represent outcomes rather than distinct service actions) were outside our scope. Our results are not generalizable to the full range of services offered by the AWWs. Future studies should examine services around the full range of ICDS services.” Please see on Page 14 of 18 in track change version

---

## [Editor Report · Decision Letter 3]

Factors associated with Anganwadi Workers’ service delivery of Integrated Child Development Services (ICDS) in rural India: A cross-sectional analysis of household and community health workers’ surveys.

PONE-D-24-18758R3

Dear Dr. Gopalakrishnan,

We’re pleased to inform you that your manuscript has been judged scientifically suitable for publication and will be formally accepted for publication once it meets all outstanding technical requirements.

Kind regards,

Rakesh Sarwal, MBBS, MPH, DrPH, ThYC

Academic Editor

PLOS ONE

---

## [Editor Report · Acceptance letter]

PONE-D-24-18758R3

PLOS ONE

Dear Dr. Gopalakrishnan,

I'm pleased to inform you that your manuscript has been deemed suitable for publication in PLOS ONE. Congratulations! Your manuscript is now being handed over to our production team.

Kind regards,

on behalf of

Dr. Rakesh Sarwal

Academic Editor

PLOS ONE